# Limit of detection of *Salmonella* ser. Enteritidis using culture-based versus culture-independent diagnostic approaches

L. M. Bradford,[1] L. Yao,[2] C. Anastasiadis,[2] A. L. Cooper,[2] B. Blais,[2] A. Deckert,[3] R. Reid-Smith,[4] C. Lau,[2] M. S. Diarra,[5] C. Carrillo,[2] A. Wong[1,6]

**ABSTRACT** To prevent the spread of foodborne illnesses, the presence of pathogens in the food chain is monitored by government agencies and food producers. The culture-based methods currently employed are sensitive but time- and labor-intensive, leading to increasing interest in exploring culture-independent diagnostic tests (CIDTs) for pathogen detection. However, few studies quantify the relative sensitivity and reliability of these CIDTs compared to current approaches. To address this issue, we conducted a comparison of the limit of detection ($LOD_{50}$) for *Salmonella* between a culture-based method and three CIDTs: qPCR (targeting *invA* and *stn*), metabarcode (16S) sequencing, and shotgun metagenomic sequencing. Samples of chicken feed and chicken caecal contents were spiked with *S.* serovar Enteritidis and subjected to culture- and DNA-based detection methods. To explore the impact of non-selective enrichment on $LOD_{50}$, all samples underwent both immediate DNA extraction and overnight enrichment prior to gDNA extraction. In addition to this spike-in experiment, feed and caecal samples acquired from the field were tested with culturing, qPCR, and metabarcoding. In general, $LOD_{50}$ was comparable between qPCR and shotgun sequencing methods. Overnight microbiological enrichment resulted in an improvement in $LOD_{50}$ with up to a three-log decrease. However, *Salmonella* reads were detected in some unspiked feed samples, suggesting false-positive detection of *Salmonella*. In addition, the $LOD_{50}$ in feeds was three logs lower than in caecal contents, underscoring the impact of background microbiota on *Salmonella* detection using all methods.

**IMPORTANCE** The appeal of culture-independent diagnostic tests (CIDTs) is increased speed with lowered cost, as well as the potential to detect multiple pathogen species in a single analysis and to monitor other areas of concern such as antimicrobial resistance genes or virulence factors. This study provides quantitative data on the sensitivity of CIDTs relative to current approaches, which is essential for determining the feasibility of implementing these methods in pathogen surveillance programs.

**KEYWORDS** surveillance, detection, *Salmonella*, metagenomics, qPCR

Foodborne pathogens inflict a serious health and economic toll worldwide. In Canada, 4 million cases of foodborne illness are thought to be domestically acquired annually, with norovirus, *Clostridium perfringens*, *Campylobacter* spp, and non-typhoidal *Salmonella* the most prevalent causes of disease (1). Detection of food pathogens throughout the food supply chain is thus critical to reduce the incidence of foodborne illness. Typically, the detection of food pathogens for surveillance and for outbreak investigation relies on isolating viable organisms using highly sensitive, culture-based methods. Since most foodborne pathogenic bacteria such as salmonellae can cause illness at very low numbers (e.g., 7 CFU) (2), methods for their detection in foods should be able to determine their presence at similarly low numbers in an analytical unit (e.g.,

**Peer Reviewer** Bryan Coburn, University of Toronto, Toronto, Ontario, Canada

Address correspondence to A. Wong, alex.wong@carleton.ca, or C. Carrillo, catherine.carrillo@inspection.gc.ca.

The authors declare no conflict of interest.

See the funding table on p. 14.

1–10 CFU per 25 g sample) (3). These highly sensitive approaches are also appropriate for commodities such as feeds, where even low doses of *Salmonella* can result in poultry colonization (4). Unfortunately, culture-based approaches can be laborious and time-consuming. For example, the time from sample collection to positive culture for *Salmonella* is up to 7 days, involving 48–72 h of enrichment culture, and 48–72 h of growth on selective agar followed by biochemical testing to confirm presumptive *Salmonella* colonies (3). In recent years, there has been increasing interest in exploring culture-independent diagnostic tests (CIDTs) such as quantitative PCR (qPCR), metabarcode sequencing, and metagenome sequencing for detecting pathogens in food (5–7) and environmental samples (8, 9), and for infectious disease diagnostics in clinical settings (10–13). These methods could offer lower costs, increased speed, and the potential to detect multiple pathogens in a single analysis. In addition, metagenome sequencing can offer insights into the presence of virulence factors (14) and antimicrobial resistance genes (15). However, pivoting to use such methods is only possible if the sensitivity and reliability of CIDTs are proven to be comparable to current approaches.

The poultry production chain is a good model for evaluating novel detection and surveillance methods, such as CIDTs. A large proportion of foodborne illnesses are associated with consumption of contaminated poultry meat (16). In the USA, over 25% of foodborne outbreaks with known sources were attributed to poultry products (17). Worldwide, a majority of cases of salmonellosis and campylobacteriosis have been associated with poultry (16, 17). Poultry products are also commonly contaminated with *Staphylococcus aureus, Listeria monocytogenes, Clostridium perfringens,* and pathogenic *Escherichia coli* (18). *Salmonella* can be introduced into poultry through feeds and persist throughout the food chain, resulting in contamination of animals and subsequent fecal contamination of retail poultry products (19, 20). Given the importance of poultry as a protein source in the global food supply, pathogen reduction in this commodity could have important human health implications. To address the question of whether CIDTs are adequately sensitive for detection of pathogens in food-relevant matrices and to obtain quantitative information on the relative performance of different methods, we conducted a comparison of the limit of detection ($LOD_{50}$) for the current culture-based *Salmonella* detection method in use at the Canadian Food Inspection Agency (CFIA) and for three CIDTs (qPCR, metabarcode sequencing, and metagenomic sequencing) in samples of chicken feed and chicken caecal contents spiked with known quantities of *Salmonella*. We further assessed the use of qPCR and 16S sequencing for *Salmonella* detection in naturally contaminated caeca and feed.

## MATERIALS AND METHODS

### Caecal and feed samples

Caeca from freshly sacrificed 35-day-old Ross 708 broiler chickens were from an ongoing study at Agriculture and Agri-Food Canada (Guelph, Ontario). All experimental procedures were approved (Protocol number # No. 3521) by the institutional ethics committees on animal experimentation according to the guidelines of the Canadian Council on Animal Care. Samples of the broiler finisher feed which included corn as the principal cereal, and soya and soybean cake as protein concentrates (Aviagen, Huntsville, United States) were used for the feed experiments. Caeca were transported on ice and stored at 4°C overnight. Feed was stored at 4°C until use. Starting materials were confirmed to be *Salmonella*-free by subjecting a subset to overnight incubation in buffered peptone water (BPW), DNA extraction, and marker-gene qPCR as described below.

### Overnight *Salmonella* culture

*Salmonella enterica* ser. Enteritidis isolate CFIAFB20140150 previously isolated from raw retail poultry [accession CP133565-CP133567; Cooper et al. (21)] was used for spiking. Bacteria were revived from a glycerol stock and plated on non-selective agar. A single

isolated colony was selected and inoculated into 5 mL buffered peptone water (BPW; Oxoid), and incubated for 24 h at 37°C with 150 rpm shaking. Previous tests of overnight cultures suggested this should result in growth to $2.5 \times 10^9$ CFU/mL. Overnight cultures were diluted in a 10X series in glucose-free M9 minimal medium (see supplementary methods), and these dilutions were used for spiking and for enumeration *via* either dropping or spreading on non-selective agar followed by overnight incubation at 37°C. Expected vs actual CFU spiked in are shown in Tables S6 and S7.

## Spiking procedure

### Caecal contents

Chicken caecal contents were "milked" into Petri dishes using sterile gloves. Sterile scoops were used to transfer 0.25 g to screw-cap tubes and 1 g to pre-dispensed 9 mL falcon tubes of BPW. Screw-cap and falcon tubes containing caecal content were spiked with between 4 and 10 µL of the appropriate dilution of the *Salmonella* ser. Enteritidis culture. Spiked caecal contents in screw-cap tubes were stored at −80°C prior to DNA extraction. For microbiological enrichment according to MFHPB-20 (3) (See Supplementary Methods), spiked caecal contents in BPW were incubated for 21 h at 35°C with 100 rpm shaking.

### Feed

For direct extractions, 10 g portions of feed was added to a filtered stomacher bag (Nasco Sampling/Whirl-Pak, United States), to which 20 mL BPW was added. The sample was homogenized using a stomacher (Interscience Laboratories, United States) for 1 min at 230 rpm. Approximately 10 mL of liquid was recovered from each sample. Samples were subjected to a low-speed spin ($500 \times g$ for 5 min) to remove eukaryotic cells. After the transfer of supernatant to a new falcon tube, samples were subjected to a high-speed spin ($11,000 \times g$ for 5 min) to pellet bacterial cells. Supernatants were discarded and the pellet was resuspended in 0.1 mL of BPW. The appropriate number of *Salmonella* cells was then added (Table S7).

For microbiological enrichments, 10 g portions of feed was added to a filtered stomacher bag, to which 90 mL of BPW was added. The sample was homogenized as described above, then spiked with 1 mL containing the appropriate dilution of *Salmonella* cells (Table S7). Samples were incubated for 20 h at 37°C.

## Growth in selective broths and agar

Recovery of *Salmonella* through secondary enrichment and growth on differential/selective agars was conducted as described in MFHPB-20 (3) (See Supplementary Methods). From the BPW enrichment, 1 mL was added to 9 mL of Tetrathionate Brilliant Green (TBG; Becton, Dickinson and Company, New Jersey, USA) broth and 0.1 mL to 9 mL of Rappaport-Vassiliadis Soya Peptone (RVS; Oxoid) broth. Inoculated TBG and RVS were incubated for 24 h at 42.5°C with 100 rpm shaking. Broths were then vortexed briefly and streaked onto Brilliant Green Sulfa (BGS; Becton, Dickinson and Company) agar and Brilliance *Salmonella* agar (Becton, Dickinson and Company) plates using 10 µL loops. Plates were incubated for 24 h at 35°C then examined for colonies indicative of *Salmonella*.

Suspected *Salmonella* colonies were confirmed using colony PCR. For caecal content samples, colonies were picked into 100 µL TE buffer, which was heated to 100°C for 10 min and then cooled to 20°C. Boiling prep material was used as a template for qPCRs. Reaction and temperature profiles are described in the qPCR section below. For feed samples, presumptive *Salmonella* colonies were confirmed by PCR amplification of the *invA* gene (Table S1). Each 25 µL reaction contained 1× GoTaq Colourless Master Mix (Promega, United States) and 0.3 µM Primers (invA_1869F, invA_1999R). Colony material was transferred directly into the PCR mix and was patched onto brain-heart infusion agar. PCR cycling conditions were as follows: denaturation at 95°C for 2 min, followed by

40 cycles of 95°C for 30 s, 60°C for 30 s, 72°C for 30 s, followed by a final extension at 72°C for 5 min. PCR products were visualized by capillary electrophoresis using a QIAxcel DNA high-resolution gel cartridge on a QIAxcel instrument (Qiagen, Toronto, Canada), according to manufacturer's instructions.

## DNA extraction

DNA extraction was performed using the DNeasy PowerSoil Pro Kit (Qiagen, Toronto, Canada) according to kit protocols. For extraction from enriched caecal samples, the remaining volume of BPW enrichments was centrifuged at $500 \times g$ for 5 min to pellet solids. Two milliliters of supernatant was centrifuged at $14,000 \times g$ for 5 min and the cell pellet was transferred to a PowerBead pro tube. For directly extracted samples, frozen spiked caecal content was thawed and beads from a PowerBead pro tube were added to the screw-cap tubes. For enriched feed samples, 10 mL of enrichments was centrifuged at $500 \times g$ for 5 min to pellet solids. The supernatant was transferred to a new tube and centrifuged at $14,000 \times g$ for 5 min and the cell pellet was transferred to a PowerBead pro tube. For direct extractions from feed, the spiked cell pellets were transferred to PowerBead pro tubes. DNA was eluted in 100 µL of elution buffer and quantified with PicoGreen (Thermo Fisher Scientific, Canada) according to the manufacturers' recommendations.

## Detection of marker genes by quantitative PCR

Detection of *Salmonella* based on the presence of marker genes *invA* and *stn* was performed by multiplex qPCR. Each reaction contained 12.5 µL of Roche FastStart Essential DNA Probes Master (Sigma-Aldrich, Oakville, Canada), 0.4 µM each *invA* primer, 0.3 µM each *stn* primer, 0.2 µM each probe (Table S1), 2.5 µL of DNA template, and water to a total volume of 50 µL. Cycling conditions are given in Table S2. The DNA template per reaction was 935 ng for caecal content samples, 3.75 ng for enriched feed samples, and 24 ng for unenriched feed samples. DNA concentrations were chosen based on standardization to the lowest sample concentration within a given group, and DNA input for enriched feed samples was further diluted to prevent overloaded reactions. Non-template controls received 2.5 µL PCR-grade water instead of DNA template. qPCRs were performed in triplicate. Duplicate standard curves in $10\times$ dilution series from $10^6$ to 1 genome copies per µL were run on each qPCR plate. The qPCR was performed on a Bio-Rad CFX Opus 96 Real-Time PCR System (Bio-Rad Laboratories Ltd., Mississauga, Canada) using the following temperature program: 95°C for 5 min, followed by 45 cycles of 95°C denaturing for 10 s, 58°C annealing for 15 s, 72°C extension for 10 s, and a final cooling step of 37°C for 30 s. Two different cycle thresholds were established for determining positivity for Salmonella: 40 cycles, based on the lack of any amplifications in no-template controls, and a more stringent setting of 35 cycles.

## Sequencing

Samples were selected for 16S and shotgun sequencing based on the results of culture-dependent and qPCR tests (Tables S6 and S7). Primers 16 S-F_341F and 16 S-R_785R [Table S1, Klindworth et al. (22)] were used to amplify the 16S V3-V4 variable region (Tables S3 and S4). Amplicon sequencing libraries were prepared according to the 16S Metagenomic Sequencing Library Preparation protocol (23) and sequenced using the MiSeq reagent kit v3 (600-cycle) for $2 \times 300$ bp paired-end sequences targeting an output of 100,000 raw reads per sample. Shotgun sequencing libraries were prepared using the Lucigen NxSeq AmpFREE Low DNA Library Kit (VWR International, Radnor, USA), and sequenced with PE150 on an Illumina NovaSeq6000, targeting an output of 50,000,000 raw reads. The actual read-depth per sample is indicated in Table S9.

## Bioinformatic analysis

### 16S

Analysis of 16S sequence data was performed in QIIME2 v2022.11 (24). Primers were removed with cutadapt using anchored forward and reverse sequences, with –p-match-read-wildcards –p-match-adapter-wildcards to account for variations in degenerate primer sequences. Untrimmed reads were discarded. Trimmed reads were denoised with DADA2(25). V3-V4 amplicons were denoised with truncation at base 260 on the forward read and 190 on the reverse read, then merged with a minimum overlap of 12 nt. Representative reads were classified using the q2-feature-classifier plugin and a Naive Bayes classifier trained on the 341–785 regions of the silva 138 database(26). Following classification, mitochondria, and chloroplast ASVs were removed using the filter-table plugin. QIIME2 output files were imported into R 4.2.3 (27) using the qiime2r package (28) and results were visualized using the phyloseq package (29) .

### Shotgun

Shotgun sequencing data sets were analyzed according to the pipeline established by Bradford et al. (30). Custom workflows were made in Snakemake (31). Briefly, reads were trimmed and quality-selected with Trimmomatic (32) using the parameters minlength 36, sliding window 4:20. All passing reads, whether paired and unpaired (forward or reverse), were retained for the best chance of *Salmonella* detection. For caecal content samples, host reads were removed by classifying passing reads with Kraken 2 (33) against a custom-made Kraken 2 database made using the *Gallus gallus* reference genome from NCBI (GRCg6a; GenBank accession GCA_000002315.5). For feed samples, reads were classified against the Kraken 2 plant database. Details on these databases can be found in the supplementary material. Reads matching the host database were removed using the filterbyname function of BBMAP (34), producing quality-controlled, host-free data sets. These reads were then classified using Kraken 2, with confidence set at 0.25, using a bacteria database downloaded using the kraken2-build command on Oct 28, 2021. All reads classified as members of the *Salmonella* genus were extracted using the filterbyname function of BBMAP. The blastx function from the Blast suite (35, 36) was used to compare putative *Salmonella* reads against a blast-formatted database of *Salmonella* "species"-specific regions from reference (37). Samples with reads that were called *Salmonella* by Kraken 2 and then passed this confirmation step are considered positive for *Salmonella*.

Reads in the unspiked (negative control) feed samples which were identified as *Salmonella*-derived *via* this pipeline were tested against the NCBI-nt database *via* the web interface. Megablast was used with default settings, excluding results from *Salmonella* (taxid:590), using the nt database posted on April 23, 2023.

## Enrichment broth dilution test

It is possible that the carrying capacity of BPW was quickly reached in caecal spiking experiments due to the high bacterial load. This would limit the possible number of divisions of *Salmonella* spiked into the broth. To determine whether dilution of the caecal contents can decrease the $LOD_{50}$ of *Salmonella*, a dilution series was conducted using 10 additional caeca obtained from Agriculture and Agri-Food Canada (Guelph, Ontario). Contents from 10 caeca were mixed and split among 16 tubes (Fig. S4). Tubes were spiked with 0 (unspiked control), 3.5, 35, or $3.5 \times 10^6$ (positive control) CFU of *Salmonella* enterica ser. Enteritidis isolate CFIAFB20140150 grown in BPW, as above. Each tube was then diluted 1:10 until the $10^3$ dilution was reached (Fig. S4). After overnight incubation, DNA was extracted using the DNeasy PowerSoil Pro Kit (Qiagen, Toronto, Canada) according to kit protocols, as above. Detection of *Salmonella* based on the presence of marker gene *invA* was performed as described above.

## Limit of detection calculations

$LOD_{50}$ of each method and condition combination was calculated according to Wilrich and Wilrich (38) using the tool provided at https://www.wiwiss.fu-berlin.de/fachbereich/vwl/iso/ehemalige/wilrich/index.html.

## Proof of concept experiment

Feed and chicken caeca were sent to laboratories at the CFIA and the Public Health Agency of Canada (PHAC) for *Salmonella* testing as part of their ongoing monitoring programs. These samples underwent culture-based detection following the MFHPB-20 protocol, and aliquots of the non-selectively enriched material were provided to us for DNA extraction and testing *via* CIDTs. DNA extraction, multiplex qPCR, and sequencing of the V3-V4 regions of the 16S rRNA gene were performed as described above. In total, 56 caeca samples and 48 feed samples were tested.

## RESULTS

We compared the limit of detection ($LOD_{50}$) of enrichment-culture-based *Salmonella* detection methodology against three CIDTs: qPCR, 16S sequencing, and metagenomic sequencing. We spiked two matrix types (chicken caecal contents and chicken feed) with known quantities of *S*. Enteritidis. For the CIDTs, all samples underwent immediate DNA extraction and an overnight enrichment incubation in non-selective media to investigate the impact of this enrichment step.

### Detection is strongly influenced by matrix

Across all methods and enrichment conditions, *Salmonella* could be detected at much lower spike-in levels in feed samples, which have low microbial abundance, than in caecal contents. The lowest $LOD_{50}$ in feed samples was 0.047 CFU/g (*via* culturing), compared to 50 CFU/g for caecal contents (*via* post-enrichment qPCR) (Fig. 1). *Salmonella* was not detected in caecal contents *via* 16S sequencing (median of 110,000 reads per sample, Table S9), regardless of enrichment condition.

### Enrichment enhances detection

In the absence of enrichment, CIDTs had considerably worse $LOD_{50}$ than traditional, culture-based testing (Fig. 1). In caeca, shotgun metagenomics (>42 million reads, Table S9) and qPCR with a Cq cutoff of 40 had $LOD_{50}$ approximately 2-log higher than culture-based detection; the use of a Cq cutoff of 35 provided an improvement of 1-log for qPCR. Lack of sensitivity was more pronounced in feed, where 16S, qPCR-35, and metagenomics had $LOD_{50}$ 3-log higher than culturing.

The first step of the culture-dependent method is an overnight incubation in BPW. To evaluate the impact of this initial incubation on test sensitivity, DNA was extracted directly from spiked samples ("unenriched") and from the BPW post-incubation ("enriched"), and these DNA extracts were used for CIDTs. The majority of reads within the shotgun sequencing data sets from unenriched feed came from plants, reducing the usable data; in contrast, plant-derived reads were a tiny proportion in the enriched feed data sets (Fig. S3). Although BPW is not selective for *Salmonella*, enrichment lowered the $LOD_{50}$ in all methods in which both conditions were tested. The $LOD_{50}$ of CIDTs using DNA extracted directly from caecal contents was particularly high, at $1.7 \times 10^3$ CFU/g for qPCR (40 cycle threshold) and $1.8 \times 10^4$ CFU/g for metagenomics *via* shotgun sequencing. With enrichment, the $LOD_{50}$ of these methods dropped to 50 and 283 CFU/g, respectively. The effect was even more pronounced in feed samples, where, for example, $LOD_{50}$ of qPCR was 21.7 CFU/g without enrichment but 0.074 CFU/g with enrichment (Fig. 1).

Enrichment was performed with 9 mL of BPW to 1 g of material as described in the culture-detection protocol (3). Diluting caecal contents to raise the BPW:material ratio

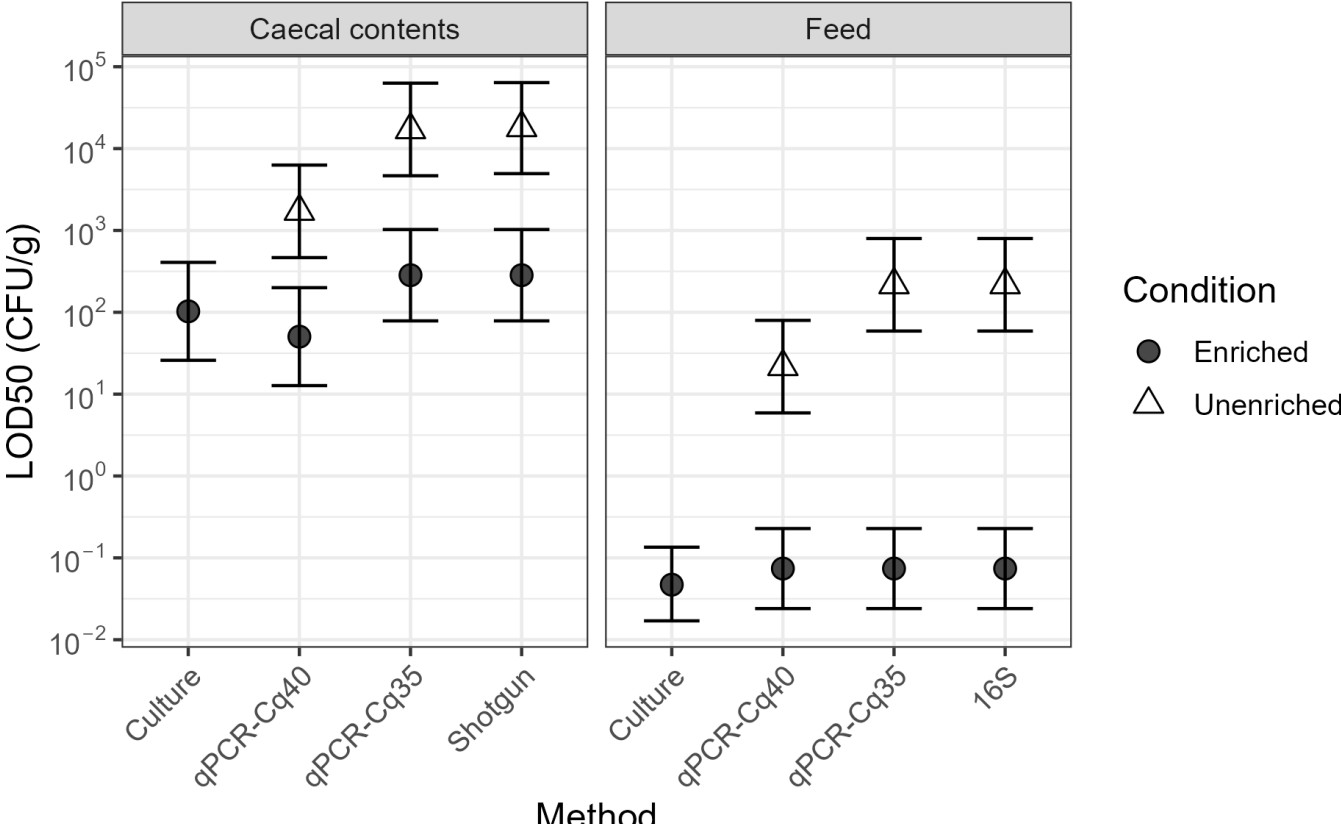

**FIG 1** Limits of detection for the methods and conditions tested according to the log-log model by Wilrich and Wilrich (38). Note that no *Salmonella* was detected in caecal contents by 16S sequencing, and $LOD_{50}$ could not be calculated for shotgun sequencing analysis of feed samples because all samples were positive. Calculations assume no *Salmonella* was detected in negative controls. 16S represents V3-V4 amplicon sequencing. qPCR-Cq40 and -Cq35 represent qPCR with Cq cutoffs of 40 and 35 cycles, respectively. Error bars show 95% confidence intervals.

improved detection, as shown with qPCR-based detection of the *invA* gene (Fig. S4). Of the six replicate samples spiked with 10 CFU/g *Salmonella* in this dilution experiment, *invA* could be detected in just one at the 9:1 ratio, in three replicates after a 10× dilution, and in all six replicates after a 100× dilution (Fig. S4).

## Enrichment has varying effects on community composition

Sequencing of 16S rRNA shows that overnight enrichment in BPW had a noticeable effect on the community composition of feed samples (Fig. 2; Fig. S1). The Enterobacteriaceae family, to which *Salmonella* belongs, was only a small proportion of the community prior to enrichment but rose to >50% post-enrichment, concurrent with a drop in alpha diversity (Fig. S2). Multiple genera within the Enterobacteriaceae greatly increased their proportion of the community during enrichment, including potentially pathogenic *Citrobacter*, *Klebsiella*, *Escherichia-Shigella*, and *Salmonella* (Fig. 2). The Actinobacter and Bacilli classes decreased in abundance, and Clostridia sequences appeared in a few feed samples following enrichment. Conversely, the overall community composition in caecal content samples showed little change (Fig. S1), and diversity dropped only slightly in enriched vs unenriched samples (Fig. S2). Enterobacteriaceae were ≤2.5% of the unenriched community and rose to 5%–13% of communities post-enrichment, but the majority of Enterobacteriaceae sequences belonged to the *Escherichia-Shigella* genus, as defined by the Silva v138.1 database (26). Sequences representing *Salmonella* were not found in any of the caecal samples selected for 16S sequencing. The most abundant class in the caecal contents was Clostridia, which comprised 89%–97% of unenriched and 77%–93% of enriched caecal communities (Fig. S1). Clostridia families Lachnospiraceae

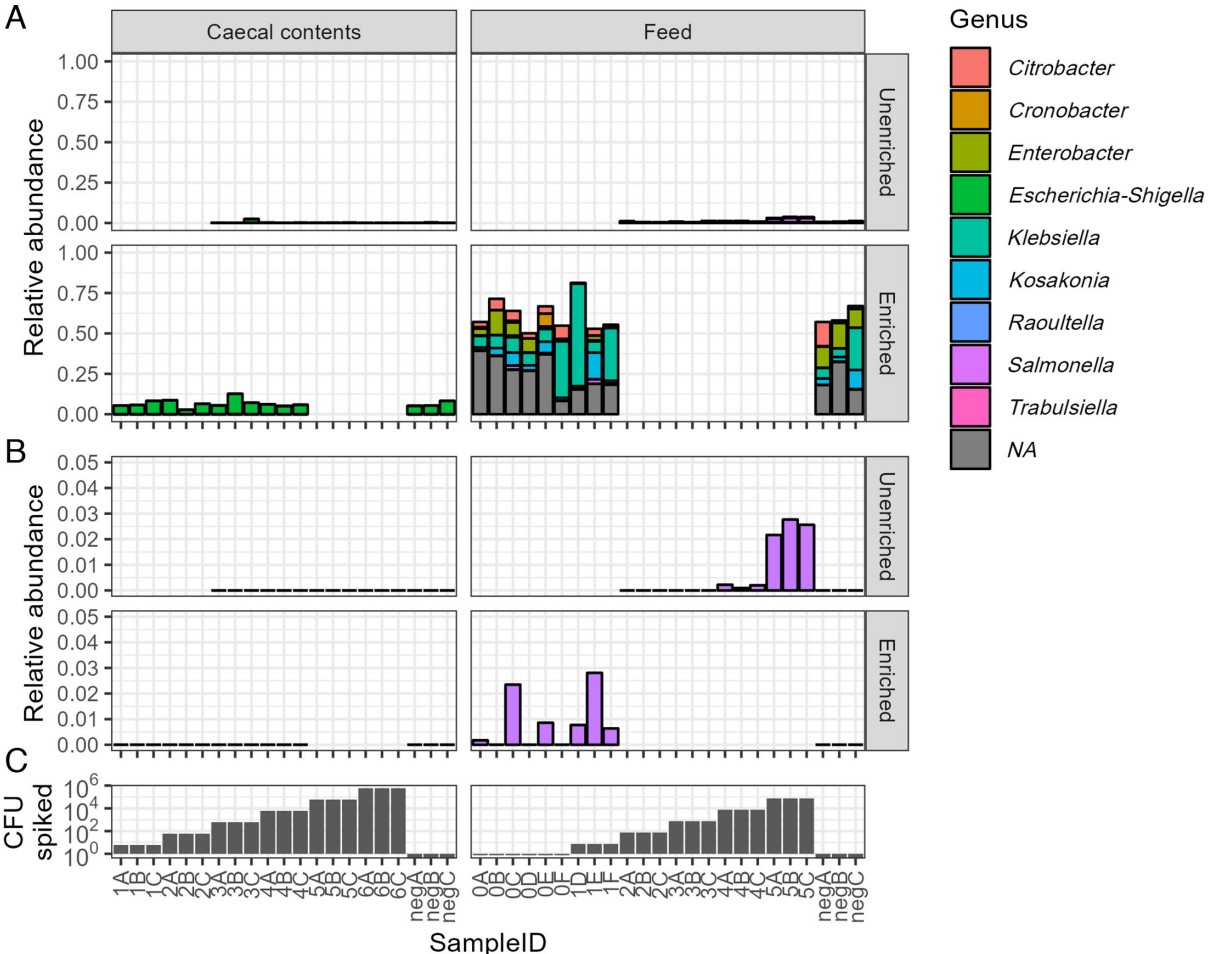

**FIG 2** (A) Relative abundance of genera in the Enterobacteriaceae family according to sequencing of the 16S V3-V4 region. Colors indicate assigned genus, with "NA" indicating sequences that could not be assigned below the family level. (B) Zoomed-in view showing only the *Salmonella* genus abundance from V3-V4 sequencing. Note the scale of the y-axis. Blank areas are shown for samples that were not sequenced. (C) The number of *Salmonella* Enteritidis CFU spiked into samples in the above panels.

and Ruminococcaeceae were 3%–28% and 4%–14 % of the total communities, respectively.

## Possible false positives for *Salmonella* in feed

Evidence of *Salmonella* was not found in unspiked feed samples *via* culturing or 16S rRNA analysis. However, one gene targeted by the multiplex qPCR (*invA*) amplified with high Cq values in two of the three enriched unspiked feed samples (Fig. 3). According to the draft protocol, amplification of either target indicates that a sample may be positive for *Salmonella*. All three enriched unspiked feed samples, as well as two of three unenriched unspiked feed samples, were found to contain shotgun sequencing reads classified as *Salmonella*-derived according to our analytical pipeline (Table S8). We carried out further investigations to determine whether these samples were in fact contaminated with *Salmonella*, or if they represent false positives. We were able to isolate and sequence colonies of *Citrobacter* species from additional feed samples and found that some sequencing reads were considered to have come from *Salmonella* when tested with our shotgun sequencing pipeline (see Supplementary Methods). These *Citrobacter* isolates, however, do not contain the *invA* gene that is tested for with qPCR.

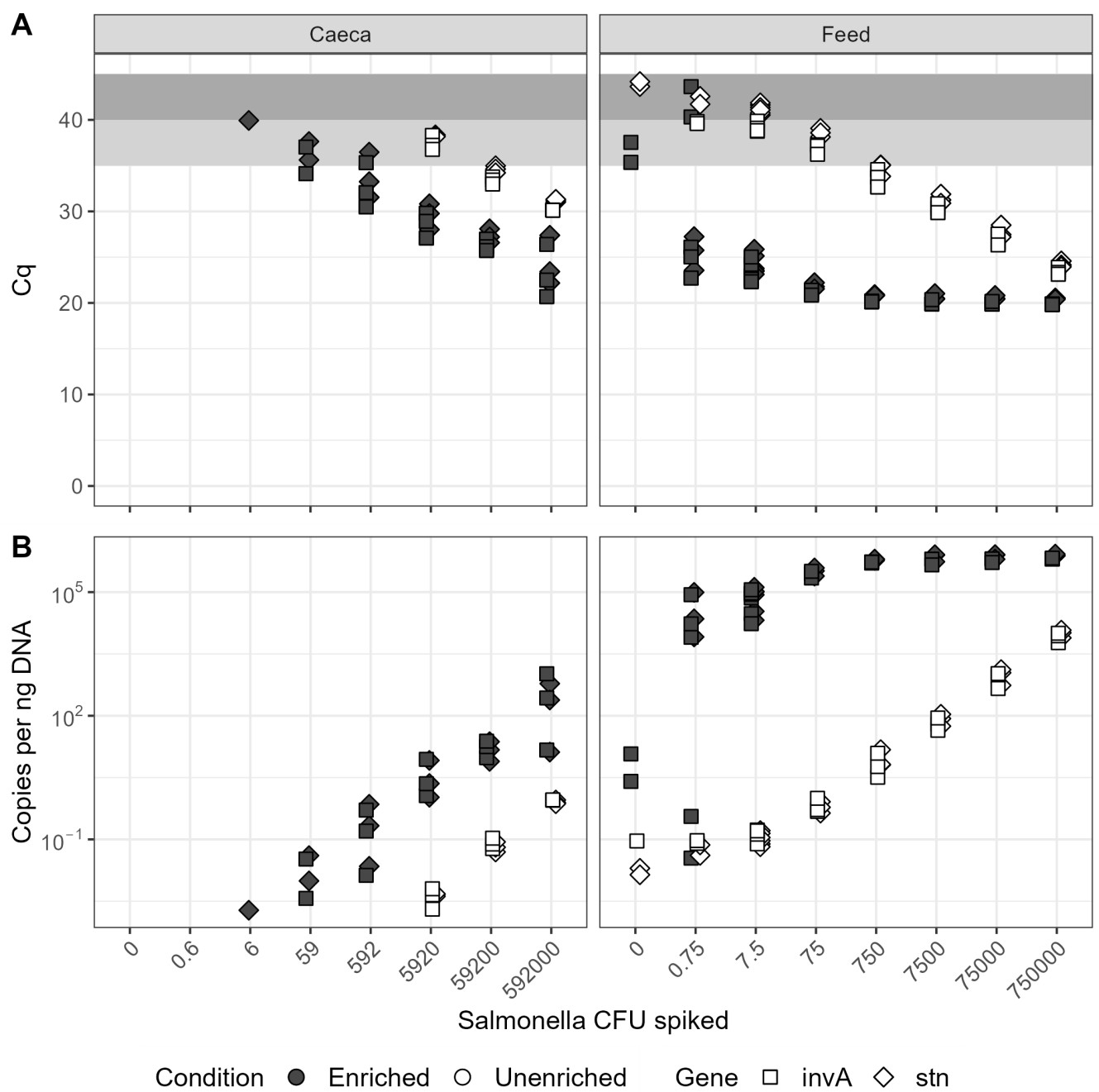

**FIG 3** Detection of *Salmonella* marker genes *via* multiplex quantitative PCR. (A) Cq values. Samples with data points in the dark gray zone above 40 cycles are considered negative; samples with data points in the light gray zone between 35 and 40 cycles may be interpreted as positive; samples with data points below 35 are definitely positive. (B) Gene copies per ng of input DNA, as calculated using standard curves. Y-axis is in log scale.

## qPCR-based detection is comparable to culturing in naturally contaminated samples

Following the spike-in experiments, a proof-of-concept experiment was performed on chicken feed and caecal contents acquired by the Canadian Food Inspection Agency and the Public Health Agency of Canada as part of their food safety monitoring programs. Culture-based testing was performed by these government agencies, and the post-enrichment material was sent to us for DNA extraction and testing by CIDTs. There was a very strong concordance between detection by culturing and by multiplex qPCR. When

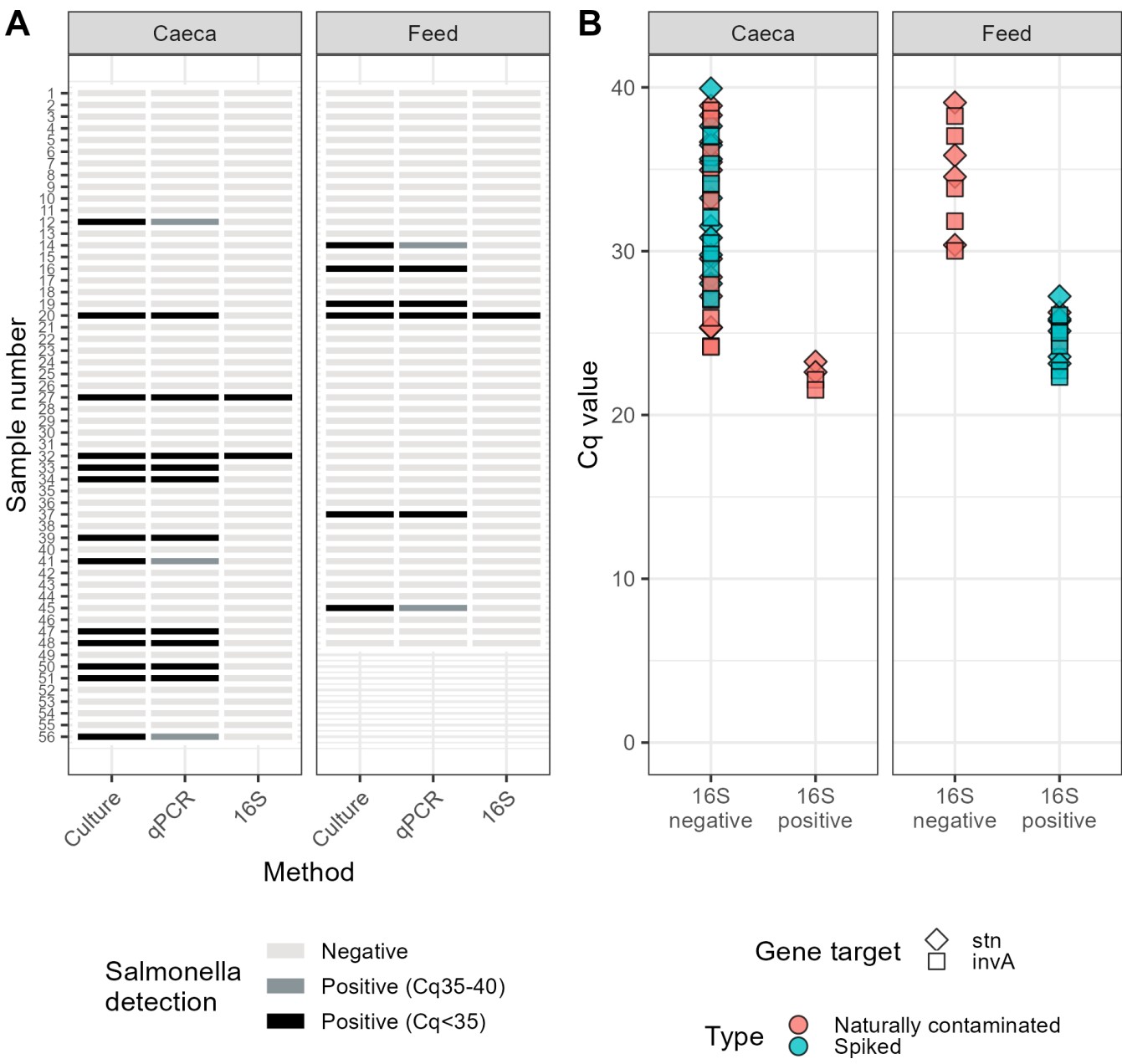

**FIG 4** (A) Results of *Salmonella* detection by culturing and CIDTs on enriched natural samples. (B) Comparison of Cq values from qPCR against positive/negative detection of *Salmonella via* sequencing of the 16S V3-V4 variable regions. Separate Cq values are plotted for the two gene targets in the multiplex qPCR assay. Results shown are from enriched samples that showed amplification in qPCRs and which underwent 16S sequencing.

qPCR positivity is defined as Cq values < 40, detection results were identical. If qPCR positivity is set more stringently with a 35 Cq threshold, 14 of the 19 culture-positive samples were found to be positive by qPCR. Detection *via* sequencing of the 16S rRNA V3-V4 regions was much less sensitive in these samples, with only one feed and two caecal samples determined to be positive for *Salmonella via* this method (Fig. 4). The samples found to be positive by 16S sequencing had low Cq values in the multiplex qPCR assay (Fig. 4).

## DISCUSSION

The primary question driving this investigation was whether various CIDTs have sufficient sensitivity and reliability to be used in food safety applications. In addition, we aimed to quantify differences in sensitivity between approaches. To address this, we systematically compared limits of detection ($LOD_{50}$) for current enrichment-culture based methodology against three CIDTs. We focused on *Salmonella* as a model pathogen, and on matrices relevant to poultry production: chicken feed (low bacterial load) and chicken caecal contents (high bacterial load). Within each matrix, we found the $LOD_{50}$ for CIDTs to be equivalent to that of the culture-dependent method when using DNA from material that underwent an overnight enrichment in non-selective broth (Fig. 1). Testing DNA extracted directly from *Salmonella*-spiked matrices yielded a higher $LOD_{50}$ in every case. Culturing approaches are known to be an effective method for increasing sensitivity, especially for the detection of small numbers of pathogenic bacteria in complex food matrices. Although enrichment is time-consuming, it is essential for detection sensitivity using CIDTs, as has been found for *Salmonella* (39, 40) and other bacterial pathogens in food matrices (41). There was good concordance between detection *via* culturing and multiplex qPCR on enriched materials, as has been seen with various qPCR methods (42–45). Although culturing, qPCR, and sequencing of the 16S rRNA V3-V4 region had equivalent LODs when tested on spiked enriched feed samples, only qPCR was able to match culturing results when used on naturally contaminated samples. Sequencing depth and quality were well-matched between these two investigations. Samples in which 16S sequencing could detect *Salmonella* were those with lower Cq values in qPCR analysis, indicating that a higher proportion of *Salmonella* DNA within the samples was needed for 16S detection with the method and sequencing depth we used. Reduced relative proportions of *Salmonella* in enrichment cultures derived from naturally contaminated samples are likely indicative of an extended lag time for *Salmonella* growth, attributed to damage to the organism due to environmental stress conditions. In addition, only one strain of *Salmonella* ser. Enteritidis was used in the spike-in portion of this study. Different strains and serovars may have variable growth kinetics in enrichment culture (46).

Some of the positive results obtained from the *Salmonella* qPCR assays had Cq values that were higher than 35 cycles. Interpretation of high Cq values may be complicated as these may represent false-positive results (47). High Cq values could be generated by degradation of probes, contamination, or by non-specific amplification of nucleic acids present in complex samples. In a diagnostic laboratory, enrichments that were qPCR positive, but with high Cq values may be further investigated by increasing the amount of sample (e.g., gDNA) loaded, or by trying to recover target organisms but these results on their own would not be conclusive. In this study, we observed "true positives" with Cq values of 40 cycles; however, some of the unspiked feed samples had a signal at this threshold. Ultimately, further evaluation of the method is needed to empirically determine reliable Cq cutoffs in a variety of matrices. In our study, we tried to maximize the amount of the gDNA sample loaded in the PCR assay to increase the relative proportion of the sample being used in the assay, particularly for the direct extraction from spiked samples. Genomic DNA from the samples was eluted into 100 µL of liquid; therefore, each qPCR assay included about 2.5% of the total sample. Total gDNA extracted from caecal contents was much higher than for feed, resulting in the use of almost 1 µg gDNA/assay for caecal samples. Further dilution to normalize feed and caecal concentrations would have significantly decreased the proportion of the sample loaded in the assay, which would have consequently impacted $LOD_{50}$.

All methods had very low $LOD_{50}$ (0.047–0.074 CFU/g) in enriched feed samples, although unenriched $LOD_{50}$ varied. This can likely be attributed to the fact that *Salmonella* cells spiked into feed were unstressed and readily viable, having just been grown in an overnight culture in rich broth. Other microorganisms on the feed had, conversely, been subsisting on dry feed material at cool (4°C) temperatures. The goal of non-selective enrichment is to allow recovery of stressed or injured cells, but it is

easy to imagine that healthy *Salmonella* enjoyed a competitive advantage over the feed microbiome in this environment, thus artificially decreasing post-enrichment LODs. For this study, we elected to forgo the stressing procedures that would typically be used in a method validation study to avoid complications associated with variability introduced by this procedure. The $LOD_{50}$ for stressed cells would likely be somewhat higher than observed here. Caecal contents, on the other hand, were freshly harvested from chickens and processed after a single night of storage at 4°C, thus minimizing stress on the resident microbiota. The majority of the caecal content community, both with and without enrichment, belongs to the Clostridia class (Fig. S1), which are common constituents of the gastrointestinal tracts of omnivorous, warm-blooded animals (48). The abundance of members of this class is consistent with surveys of chicken caecal communities (48). All Clostridia are obligate anaerobes (49), which would not be expected to maintain an overwhelming presence after enrichment in an oxic environment. One possible explanation is that, due to the high biomass in caecal content, the carrying capacity of the broth was quickly reached with very little opportunity for the growth of aerobes. Results of an experiment in which caecal contents were serially diluted in BPW before overnight enrichment support this hypothesis, with improved qPCR-based detection in samples with higher BPW:caecal content ratios during enrichment (Fig. S4).

The relatively high $LOD_{50}$ for *Salmonella* in caecal contents has implications for monitoring schemes that rely on testing these materials, notably the National Microbiological Baseline Study in Broiler Chicken December 2012 (50). That study suspended chicken caecal contents in a 1:4 (wt/wt) ratio with BPW, then screened using the BAX PCR system (Hygenia, Mississauga, Canada), with presumptive positives enumerated by Most Probably Number (MPN) culturing. They found that 25.6% of the caecal samples tested were positive for *Salmonella*, with 65% of those positives enumerated at >110 MPN/g. However, our results suggest that the positivity rate may have been higher, but hidden by the inability of *Salmonella* to grow sufficiently during enrichment. Our findings may also have implications for other studies and monitoring schemes that test for pathogens in high biomass backgrounds such as probiotic preparations and fermented consumable products (51, 52).

While buffered peptone water (BPW) is considered a non-selective medium, we found clear evidence that overnight growth in BPW favors the growth of some taxa to the exclusion of others. Non-selective enrichment of feed caused profound changes in the bacterial community compositions. Previous studies on non-selective enrichment (using BPW or Universal Pre-enrichment Broth, UPB) of various food products saw a decrease in the proportion of Proteobacteria (which includes *Salmonella*) and an increase in Firmicutes, with varying results for Actinobacteriota (53–55). Conversely, non-selective enrichment in our experiment caused an increased proportion of Proteobacteria, a decrease in Firmicutes, and the near-disappearance of the Actinobacteriota phylum. The Proteobacteria phylum consisted mostly of members of the Enterobacteriaceae family, including *Citrobacter*, *Klebsiella*, *Escherichia-Shigella*, and *Salmonella* genera. There is thus a need for further work on the effects of enrichment on the microbial communities of different commodities.

Amplification of the *invA* gene during qPCR and detection of putatively *Salmonella*-derived shotgun sequencing reads in unspiked feed sample controls suggest that *Salmonella* DNA may have been present. This does not guarantee the presence of viable cells; indeed, the inability of CIDTs to distinguish between viable cells and lingering DNA is a known downfall of these methods (56, 57). It is also possible that signals were generated from nonspecific products generated in these complex samples (58). The number of reads identified as coming from *Salmonella* was higher in enriched samples than in their unenriched counterparts, which could indicate the growth of viable cells. The more likely explanation is that these reads are false positives due to the presence of related organisms. We previously isolated a *Citrobacter werkmanii* from feed that contains sequences matching those found in the unspiked feed controls and have since

isolated multiple *Citrobacter* colonies from feed used in this experiment with sequences that are attributed to *Salmonella* in our bioinformatic pipeline. Characterization of these isolates is ongoing. *Citrobacter* spp. are closely related to *Salmonella* (59) and have been shown to cause false positives during food testing (60, 61). The genome of the previously isolated *Citrobacter* has not been uploaded to NCBI or other databases, so it was not available during the determination of the *Salmonella* species-specific regions used during bioinformatic analysis (37), although shotgun reads simulated from its genome were tested during pipeline development and did not result in false *Salmonella* hits (30). Read classification in metagenomic analysis relies on matching sequences to curated databases (62). Over-representation of pathogenic species in public repositories relative to commensal organisms commonly found in food and environmental species has the potential to lead to false-positive detection of pathogens as observed in this study (63). This emphasizes the need for caution when using CIDTs for food safety or in health diagnostics.

Overall, there were advantages and disadvantages observed for each of the CIDTs investigated. Shotgun sequencing of enrichment cultures, while slightly less sensitive than qPCR, could offer valuable insights into the microbial composition of the samples. Unlike qPCR, which is limited to detecting known targets, shotgun sequencing provides comprehensive data on the strains present in the culture. This method can reveal whether the sample contains multiple strains or just one, making it a potentially valuable addition for strain-level characterization and outbreak investigation(64). Metabarcoding based on sequencing the 16S rRNA V3-V4 region was also a reliable method for detecting *Salmonella*, offering high specificity and sensitivity in identifying this pathogen. Similar to shotgun metagenomics, this approach provides comprehensive information on the microbiota present in enrichment samples, aiding in interpreting the overall microbial context and potential interferences. Our results indicate that higher sequencing depths (e.g., greater than 100,000 reads) would be needed to improve sensitivity in some enrichments. While less practical than qPCR this method could be of value when investigating difficult food matrices.

CIDTs are promising tools for pathogen surveillance and detection in agriculture, food safety, and medicine. However, the performance of CIDTs must be systematically investigated to guide their appropriate use. Here, we show that the CIDTs tested have equivalent sensitivity to culture-based detection methods when an overnight incubation is employed, but much higher limits of detection (i.e., lower sensitivity) without this enrichment. The detection limits of all methods are clearly influenced by the matrix background, which must be considered when interpreting results from varied matrices. We also show the major downside of CIDTs, that is, the potential for false positives and lack of cultured isolates on which to perform further tests.

## ACKNOWLEDGMENTS

Thank you to the personnel at the Public Health Agency of Canada and the Canadian Food Inspection Agency who provided samples for the proof-of-concept portion of this research.

Funding for this project was provided by the Ontario Ministry of Agriculture, Food, and Rural Affairs (OMAFRA project number OAF-2020-101088).

## AUTHOR AFFILIATIONS

[1]Department of Biology, Carleton University, Ottawa, Ontario, Canada

[2]Research and Development, Ottawa Laboratory (Carling), Canadian Food Inspection Agency, Ottawa, Ontario, Canada

[3]Centre for Foodborne Environmental and Zoonotic Diseases, Public Health Agency of Canada, Guelph, Ontario, Canada

[4]Public Health Agency of Canada, Ottawa, Ontario, Canada

⁵Guelph Research and Development Centre, Agriculture and Agri-Food Canada, Guelph, Ontario, Canada
⁶Institute for Advancing Health Through Agriculture, Texas A&M University, Fort Worth, Texas, USA

**PRESENT ADDRESS**

L. M. Bradford, Environmental Health Science and Research Bureau, Health Canada, Ottawa, Ontario, Canada

**AUTHOR ORCIDs**

C. Carrillo  http://orcid.org/0000-0002-2334-8718
A. Wong  http://orcid.org/0000-0001-6249-3013

**FUNDING**

| Funder | Grant(s) | Author(s) |
|---|---|---|
| Ontario Ministry of Agriculture, Food and Rural Affairs (OMAFRA) | OAF-2020-101088 | C. Carrillo |
| | | A. Wong |

**AUTHOR CONTRIBUTIONS**

L. M. Bradford, Conceptualization, Data curation, Formal analysis, Funding acquisition, Investigation, Methodology, Project administration, Supervision, Writing – original draft, Writing – review and editing | L. Yao, Conceptualization, Data curation, Formal analysis, Investigation, Methodology, Writing – original draft, Writing – review and editing | C. Anastasiadis, Investigation | A. L. Cooper, Investigation | B. Blais, Conceptualization, Investigation, Writing – review and editing | A. Deckert, Conceptualization, Methodology, Writing – review and editing | R. Reid-Smith, Conceptualization, Methodology, Resources | C. Lau, Conceptualization, Methodology, Resources | M. S. Diarra, Conceptualization, Methodology, Resources | C. Carrillo, Conceptualization, Methodology, Resources, Supervision, Writing – review and editing | A. Wong, Conceptualization, Funding acquisition, Methodology, Project administration, Resources, Supervision, Writing – review and editing

**DATA AVAILABILITY**

The data have been deposited to NCBI (BioProject accession number PRJNA1035945). The code can be found at https://github.com/LMBradford/SalmLOD-paper.

**ETHICS APPROVAL**

All experimental procedures were approved (Protocol No. 3521) by the Institutional Ethic Committees on Animal Experimentation according to the guidelines of the Canadian Council on Animal Care.

**ADDITIONAL FILES**

The following material is available online.

Supplemental Material

**Supplemental material (Spectrum01027-24-s0001.pdf).** Tables S1 to S9; Fig. S1 to S4.

Open Peer Review

**PEER REVIEW HISTORY (review-history.pdf).** An accounting of the reviewer comments and feedback.

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
