## [Reviewer comments · Microbiology Spectrum]

Microbiology Spectrum

Limit of detection of *Salmonella* ser. Enteritidis using culture-based versus culture-independent diagnostic approaches

Lauren Bradford, Lang Yao, Chloe Anastasiadis, Ashley Cooper, Burton Blais, Anne Deckert, Richard Reid-Smith, Calvin Lau, Moussa S. Diarra, Catherine Carrillo, and Alex Wong

Corresponding Author(s): Alex Wong, Carleton University

Review Timeline:

Submission Date:	May 3, 2024
Editorial Decision:	May 29, 2024
Revision Received:	August 9, 2024
Accepted:	August 28, 2024

Editor: Vincenzina Fusco

Reviewer(s): Disclosure of reviewer identity is with reference to reviewer comments included in decision letter(s). The following individuals involved in review of your submission have agreed to reveal their identity: Bryan Coburn (Reviewer #2)

Transaction Report:

DOI: <https://doi.org/10.1128/spectrum.01027-24>

Re: Spectrum01027-24 (Limit of detection of Salmonella ser. Enteritidis using culture-based versus culture-independent diagnostic approaches)

Dear Dr. Alex Wong:

Thank you for the privilege of reviewing your work. Below you will find my comments, instructions from the Spectrum editorial office, and the reviewer comments.

Revision Guidelines

Sincerely,
Vincenzina Fusco
Editor
Microbiology Spectrum

Reviewer #1 (Comments for the Author):

This manuscript compares CIDT methods to detect Salmonella to the culture method in MFHPB-20. The sample population includes spiked and unspiked Chicken caecal contents and chicken feed, and naturally contaminated caecal and feed samples. DNA was extracted from samples prior and following a non-selective pre-enrichment in BPW. The CIDTs include qPCR of Salmonella *invA* and *stn* genes, 16S Amplicon sequencing of the V4 and V3-V4 regions of the 16srRNA gene and shotgun

metagenomic sequencing. The manuscript is well written, but the results are predictable. It is well known that culture enrichment is required to detect foodborne pathogens via CIDTs, and the authors do not acknowledge this adequately. Since the authors state that the primary question driving this investigation was whether CIDTs are as sensitive as culture methods and the answer is already known this manuscript is not adding any new or novel information to science.

Additional comments

It is not clear why the authors included the V4 16S data since it is not a good target for Salmonella detection. Here is a study that shows the V1 - V3 regions have the highest sensitivity for Salmonella and also addresses how the background flora can influence detection. And, there are more studies to look at variable region specificity differences between bacteria.

High-Resolution Microbiome Profiling for Detection and Tracking of *Salmonella enterica*.

Grim CJ, Daquigan N, Lusk Pfefer TS, Ottesen AR, White JR, Jarvis KG.

Front Microbiol. 2017 Aug 18;8:1587. doi: 10.3389/fmicb.2017.01587. eCollection 2017.

There is no information about the multiplexing strategies used for sequencing. The number of samples per sequencing run for the V3-V4 data from the MiSeq could impact the sensitivity of detection.

Did the shotgun metagenomic data support the 16S data?

The authors do not mention uploading their data to NCBI.

The authors did not spell out and thoroughly describe the MFHPB-20 method. The reference provided leads to a link to an email address.

Reviewer #2 (Comments for the Author):

This is a well conceptualized and clearly written study with a focused aim and conclusions appropriate for the methods, relevant to the question (and field) and useful for similar applications in food and other microbiologic diagnostic testing.

Thanks for writing such a clear and easy-to-understand manuscript with appropriate figures, legends and data. I had a very good understanding of the methods and findings after a single pass.

I admit I struggled a bit at first to understand what the utility of the 16S analysis was post-enrichment, and am still not certain that it is especially useful in the context of pathogen detection testing, although the data are presented clearly enough.

We sincerely appreciate the time and effort dedicated to reviewing our manuscript. We have made every effort to address each of the review comments as follows:

Reviewer #1:

1. This manuscript compares CIDT methods to detect Salmonella to the culture method in MFHPB-20. The sample population includes spiked and unspiked Chicken caecal contents and chicken feed, and naturally contaminated caecal and feed samples. DNA was extracted from samples prior and following a non-selective pre-enrichment in BPW. The CIDTs include qPCR of Salmonella invA and stn genes, 16S Amplicon sequencing of the V4 and V3-V4 regions of the 16srRNA gene and shotgun metagenomic sequencing. The manuscript is well written, but the results are predictable. It is well known that culture enrichment is required to detect foodborne pathogens via CIDTs, and the authors do not acknowledge this adequately. Since the authors state that the primary question driving this investigation was whether CIDTs are as sensitive as culture methods and the answer is already known this manuscript is not adding any new or novel information to science.

Response:

We appreciate this feedback. We agree that it is well known that enrichment would enhance detection and we have revised the manuscript to make this clear. One of the key objectives of our study was to obtain quantitative data on the degree of improvement provided by enrichment and the relative sensitivity of the different approaches. Additionally, we aimed to understand the differences in detection capabilities between samples with high microbial loads and those with low microbial backgrounds. This quantitative information is crucial for risk modelers as it offers valuable data on the limit of detection for various approaches. By providing these insights, our study contributes important quantitative benchmarks that are essential for refining and validating CIDT methodologies in the context of food safety testing. We have added statements to the introduction and discussion within the manuscript to clarify these goals and the value of the quantitative information obtained.

2. It is not clear why the authors included the V4 16S data since it is not a good target for Salmonella detection. Here is a study that shows the V1 - V3 regions have the highest sensitivity for Salmonella and also addresses how the background flora can influence detection. And, there are more studies to look at variable region specificity differences between bacteria.
High-Resolution Microbiome Profiling for Detection and Tracking of Salmonella enterica.
Grim CJ, Daquigan N, Lusk Pfefer TS, Ottesen AR, White JR, Jarvis KG.
Front Microbiol. 2017 Aug 18;8:1587. doi: 10.3389/fmicb.2017.01587. eCollection 2017.

Response:

We acknowledge that the V4 region is not the optimal target for Salmonella detection. Given this feedback and the supporting literature, we have decided to remove the V4 16S data from the manuscript.

3. There is no information about the multiplexing strategies used for sequencing. The number of samples per sequencing run for the V3-V4 data from the MiSeq could impact the sensitivity of detection.

Response:

In response to this comment, we have modified the Materials and Methods section to indicate targeted number of reads (p. 8, line 184). Actual raw read counts for each method were provided in Table S9. We now refer to this table in the results section so that readers can more easily interpret results in the context of sequencing depth. This additional information should clarify our multiplexing strategy and provide a more comprehensive understanding of the sequencing depth and its implications for detection sensitivity.

4. Did the shotgun metagenomic data support the 16S data?

Response:

We appreciate the reviewer's question regarding the support of 16S data by the shotgun metagenomic data. Since we employed a specialized pipeline focused on the specific detection of Salmonella, we did not obtain data for other species. Although it would be possible to conduct additional analyses to include other species, such analyses would not significantly enhance the findings or conclusions of our study.

5. The authors do not mention uploading their data to NCBI.

Response:

We apologize for any confusion. All of the data have indeed been deposited in BioProject under accession number PRJNA1035945, as described in the "Data Availability" section of the Materials and Methods.

6. The authors did not spell out and thoroughly describe the MFHPB-20 method. The reference provided leads to a link to an email address.

Response:

Unfortunately, Health Canada requires an email request to obtain the official MFHPB-20 method rather than providing a direct download link. We understand

this may be inconvenient, but we assure you that the method is promptly provided upon request via email. The details of the method that are pertinent to this study have been provided in the materials and methods section “Growth in selective broths and agars”. We have provided additional details to facilitate retrieval of the method in the Supplementary methods.

Reviewer #2:

7. This is a well conceptualized and clearly written study with a focused aim and conclusions appropriate for the methods, relevant to the question (and field) and useful for similar applications in food and other microbiologic diagnostic testing. Thanks for writing such a clear and easy-to-understand manuscript with appropriate figures, legends and data. I had a very good understanding of the methods and findings after a single pass.

Response:

We are grateful for the reviewer’s positive feedback and are pleased to hear that you found our study to be well-conceptualized, clearly written, and relevant to the field. Keeping it simple became increasingly difficult as we conducted follow-up studies to better understand our results. Your encouraging comments are highly appreciated.

8. I admit I struggled a bit at first to understand what the utility of the 16S analysis was post-enrichment, and am still not certain that is especially useful in the context of pathogen detection testing, although the data are presented clearly enough.

Response:

We appreciate the reviewer’s feedback regarding the utility of the 16S analysis post-enrichment. In response, we have removed the V4 16S data to reduce the emphasis on this analysis as a pathogen detection method, especially given the results of these analyses. Our intention was to use the 16S data as an example of well-validated genus markers and to assess the enrichment dynamics of the culture. Additionally, 16S sequencing could theoretically be a less expensive alternative to shotgun sequencing, making it a viable option for certain applications. We have added statements at the end of the discussion to clarify these advantages.

Re: Spectrum01027-24R1 (Limit of detection of *Salmonella* ser. Enteritidis using culture-based versus culture-independent diagnostic approaches)

Dear Dr. Alex Wong:

Your manuscript has been accepted, and I am forwarding it to the ASM production staff for publication. Your paper will first be checked to make sure all elements meet the technical requirements. ASM staff will contact you if anything needs to be revised before copyediting and production can begin. Otherwise, you will be notified when your proofs are ready to be viewed.

Sincerely,
Vincenzina Fusco
Editor
Microbiology Spectrum

Reviewer #1 (Comments for the Author):

The revisions are great. Thank you for addressing all comments and concerns so thoroughly.

Reviewer #2 (Comments for the Author):

I was reviewer #2 on the prior submission.

I find the revised submission helpful - the quantitation of the differences in sensitivity between CIDTs and culture-enriched analyses is useful when designing pathogen-detection studies, and will enable a more informed assessment of cost-workload/sensitivity tradeoffs, amongst other considerations.

The data are clear and support the conclusions with a clear description of methods that ensure the approach can be duplicated.

I have no additional concerns and believe this analysis will be useful to the field. Thanks very much for this work - it's not glamorous, but it is definitely informative and important!